## RESEARCH ARTICLE

# Imp and Chinmo are required for embryonic motor neuron axon and dendrite targeting

Katherine H. Fisher, Sen-Lin Lai and Chris Q. Doe*

## ABSTRACT

Neural progenitors generate distinct neuronal populations over time. *Drosophila* larval neural progenitors, neuroblasts (NBs), generate neuronal diversity by expressing temporal gradients of transcription factors and RNA-binding proteins, including early factors Imp and Chinmo and late factors Syp, Mamo, and Broad. These factors have been well characterized in the larval central nervous system (CNS), yet nothing is known about their expression or function in the embryonic CNS. We show that embryonic Imp is expressed in a low-to-high temporal gradient, the opposite of the larval Imp gradient. Embryonic Chinmo is expressed in all post-mitotic neurons, but not in a gradient, while the late larval factors Mamo, E93, Syp, and Broad show little embryonic expression. We show that Imp is required for Chinmo expression in postmitotic neurons, and loss of Chinmo – but not Imp – derepresses Syp. Finally, we tested whether Imp and Chinmo are required for motor neuron molecular identity or morphology. Although neither is required to specify temporal or molecular neuronal identity, both are required for axon targeting to the correct body wall muscle, and downregulating dendrite outgrowth. We conclude that temporal factors are regulated differently in embryos and larvae, and that Imp and Chinmo are required for proper neuronal axon and dendrite projections.

KEY WORDS: Neuroblast, Chinmo, Imp, Syncrip, Motor neuron, Axon, Dendrite, RNA-binding protein

## INTRODUCTION

The generation of distinct populations of neurons is an essential part of neurogenesis. Neurons with diverse function, connectivity, and morphology are important for sensation and generation of complex behaviors across the animal kingdom. Neural progenitors give rise to distinct populations of neurons throughout development. In *Drosophila* and mammals, spatial patterning of the neuroectoderm generates molecularly distinct progenitor pools (mammals) or distinct individual progenitors [*Drosophila* neuroblasts (NBs)] (Crews, 2019; Erclik et al., 2017; Guillemot, 2007; Sagner and Briscoe, 2019). Subsequently, each progenitor undergoes changes in gene expression over time, a process called temporal patterning. Temporal patterning occurs in the mammalian spinal cord, cerebral

Institute of Neuroscience, Howard Hughes Medical Institute, University of Oregon, Eugene, OR 97403, USA.

*Author for correspondence (cdoe@uoregon.edu)

K.H.F., 0000-0002-3513-6616; S.-L.L., 0000-0002-7531-283X; C.Q.D., 0000-0001-5980-8029

cortex, and retina (Mattar and Cayouette, 2015; Sagner et al., 2021; Sagner and Briscoe, 2019), and in the *Drosophila* embryonic ventral nerve cord (VNC; analogous to the vertebrate spinal cord), larval central brain, and optic lobe (Doe, 2017; El-Danaf et al., 2023).

In *Drosophila*, temporal patterning occurs via two distinct mechanisms: a cascade of transcription factors, or gradients of RNA-binding proteins. The best characterized temporal transcription factor (TTF) cascade is in the VNC, where NBs sequentially expresses Hunchback (Hb), Krüppel, Pdm1/2, Castor, and Grainy head (Doe, 2017). Due to displacement of neurons from NB divisions, early- and late-born cell type are spatially separated in the VNC cortex. Hb+ early born cells are positioned deep in the cortex and late-born Cas+ cells are located superficially near the NBs (Seroka and Doe, 2019). A comparable TTF cascade occurs within the progeny of central brain Type II NBs, called intermediate neural progenitors (INPs), which sequentially express the transcription factors Dichaete, Grainy head, and Eyeless over several rounds of cell division (Bayraktar and Doe, 2013; Homem et al., 2013; Tang et al., 2022). Finally, distinct TTFs are used in the optic lobe (El-Danaf et al., 2023). In all three regions of the CNS, the concept is the same: each TTF in the cascade specifies one or a few specific neuronal and glial cell types.

A second mechanism of temporal patterning occurs in larval central brain NBs, where opposing gradients of two RNA-binding proteins, IGF-II mRNA binding protein (Imp) and Syncrip (Syp), specify different neuronal identities based on the level of each protein (Fig. 1) (Guan et al., 2022; Liu et al., 2015). In the mushroom body, Imp and Syp proteins are expressed in opposing gradients and cross-repress each other (Liu et al., 2015). Imp is expressed early in a high-to-low temporal gradient, while Syp is expressed late in an opposing low-to-high temporal gradient. Knockout of Imp or Syp results in dramatic loss of early- or late-born mushroom body cell types, respectively (Liu et al., 2015). Post-transcriptional regulation of translation of the transcription factor Chinmo adds an additional layer of neuronal diversity. Imp positively regulates Chinmo expression, while Syp represses Chinmo expression through binding of the 5′UTR (Liu et al., 2015; Zhu et al., 2006). Furthermore, Chinmo activates expression of Mamo, creating another layer of temporal diversity by generating an intermediate mushroom body cell type (Liu et al., 2019). Similarly, in Type II central brain NBs (Bello et al., 2008; Boone and Doe, 2008; Bowman et al., 2008), Imp and Syp are also expressed in opposing gradients, and Imp is required for early-born neuron fates, while Syp is required for late-born fates (Ren et al., 2017; Syed et al., 2017). Chinmo is expressed early alongside Imp, and both are repressed by Syp midway through larval development (Ren et al., 2017). In the latter half of larval development, the expression of two transcription factors, E93 and Broad, function to specify late neuronal fates (Ren et al., 2017; Syed et al., 2017). Finally, Imp is required for proper axon/dendrite targeting in the central complex (Munroe and Doe, 2023).

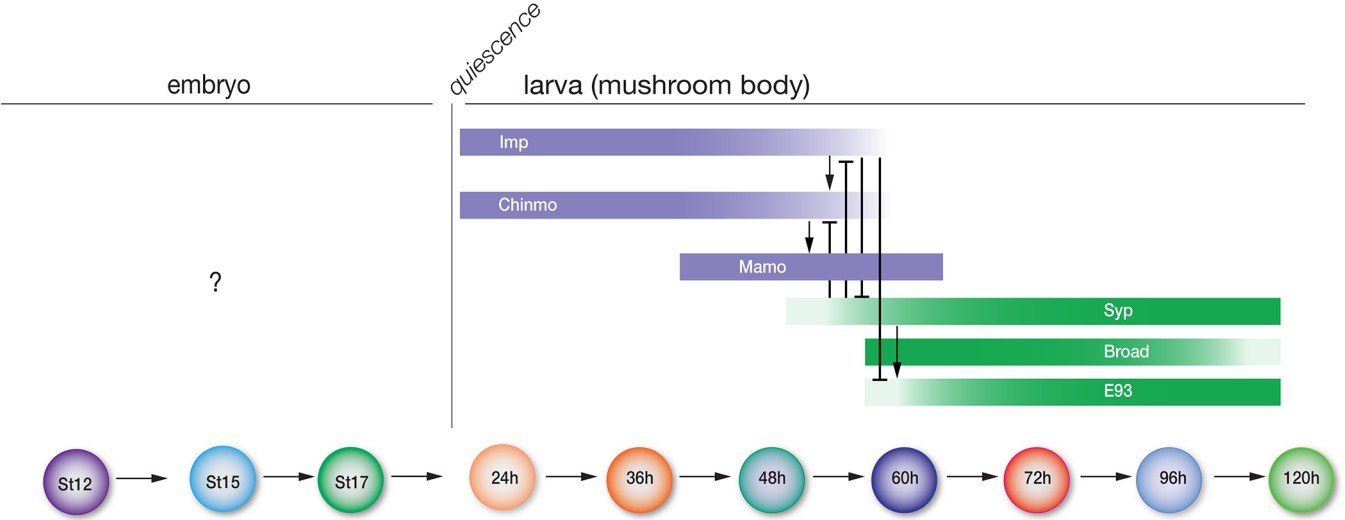

**Fig. 1. Schematic of known and unknown roles for the larval temporal factors.** Known roles of the indicated larval temporal factors over time are shown on the right; none of these factors has been investigated for a role in the embryonic ventral nerve cord (left). St, embryonic stage. h, hours after larval hatch. Timeline not to scale.

While the gene regulatory network of these factors has been thoroughly characterized in larvae, little is known about the role of these larval temporal factors – Imp, Syp, Chinmo, Mamo, Broad, E93 – in embryonic CNS development. What is known suggests that different mechanisms are used in embryonic versus larval NBs: Castor is a late TTF in the embryonic CNS (Doe, 2017), whereas it is an early TTF in larval NBs (Dillon and Doe, 2024). The major unknown questions are as follows. Are the larval factors expressed in the embryonic CNS? Do Imp/Syp form opposing gradients, as in larvae? What is the relationship between the embryonic TTF cascade and Imp/Syp expression? Do they specify neuronal molecular identity, or later aspects such as axon or dendrite morphology? Here, we address these questions, finding both similarities and differences in larval and embryonic Imp/Syp/Chinmo expression and function. Notably, we found that both Imp and Chinmo are required for proper embryonic motor neuron axon targeting to their proper muscles, and they are both required to prevent dendrites from mistargeting within the CNS neuropil.

## RESULTS
### Imp is expressed in a low-to-high temporal gradient in embryonic neurons
In larval development, Imp is expressed in a high-to-low temporal gradient in NBs, and expression levels are inherited by the daughter cells (Islam and Erclik, 2022). In the embryonic CNS, Imp expression has only been characterized at low resolution and via a GFP knock-in tagged Imp:GFP protein (Adolph et al., 2009). We used antibodies to detect Imp, Syp, and Chinmo proteins to determine their expression in embryonic CNS development. We found that Imp is detected in the cytoplasm of all embryonic NBs (Fig. 2A). Imp expression levels are similar in NBs until stage 12, at which time they show an increase in Imp levels (Fig. 2A,B). Imp also shows a low-to-high temporal gradient in post-mitotic neurons (Fig. 2C,D), although it does not form a temporal gradient in Eve+ U1-U5 motor neurons (Fig. 2E,F) or in the Eve+ EL interneurons (Fig. 2E-H) – two populations that contain both early-born and late-born neurons (Seroka and Doe, 2019; Tsuji et al., 2008; Wreden et al., 2017). Thus, Imp forms a low-to-high temporal gradient in NBs and some post-mitotic neurons. This is the opposite of the Imp high-to-low temporal gradient in larvae and was our first sign that the larval factors are expressed

differently in the embryonic CNS. In contrast, the early factor Chinmo is first detected in neurons beginning at stage 12, without forming a temporal gradient (Fig. 2C-H). The late larval factors are also expressed differently in the embryonic CNS: Broad is expressed in a subset of neurons and has higher expression in thoracic segments; and E93, Mamo, and Syp were not present in embryonic NBs or neurons (Fig. S1). Overall, we conclude Imp is expressed in a low-to-high temporal gradient in NBs and some post-mitotic neurons, whereas Chinmo is first expressed in neurons, not NBs, and does not form an expression gradient.

### The Imp temporal gradient and TTF cascade are independent of each other
In the larval central brain, there is one mechanism known to specify NB identity: gradients of RNA-binding proteins; a TTF cascade has yet to be identified (Ren et al., 2017; Syed et al., 2017). In the embryonic VNC, there are two potential mechanisms that may specify neuronal identity: the well-characterized TTF cascade (Doe, 2017) and the Imp temporal gradient described above. Here, we investigated the relationship between the two mechanisms: does the TTF cascade generate the Imp temporal gradient, or does the Imp gradient regulate the TTF cascade, or are they independent? To determine if the TTF cascade generates the Imp temporal gradient, we used *en-gal4*, which is expressed in stripes within the posterior domain of each segment, to misexpress *UAS-hb* – which is known to stall the TTF cascade at the first TTF (Isshiki et al., 2001; Pollington and Doe, 2025; Tran and Doe, 2008) – and assayed Imp levels. We found that Hb misexpression had no effect on Imp levels (Fig. S2A,B). We next asked whether Imp regulates the TTF cascade. We used *en-gal4* to misexpress *UAS-Imp* and found that Imp misexpression had no effect on progression of the TTF cascade: Hb NB expression was on at stage 10 and off at stage 12, similar to controls (Fig. S2C-F). We conclude that the Imp temporal gradient and the TTF cascade are generated independently.

### Embryonic cross-regulation of Chinmo, Imp, and Syp is different from larval cross-regulation
There is one mechanism known to specify temporal identity in larvae: Chinmo and Imp show cross-regulation (Fig. 1, right) (Guan

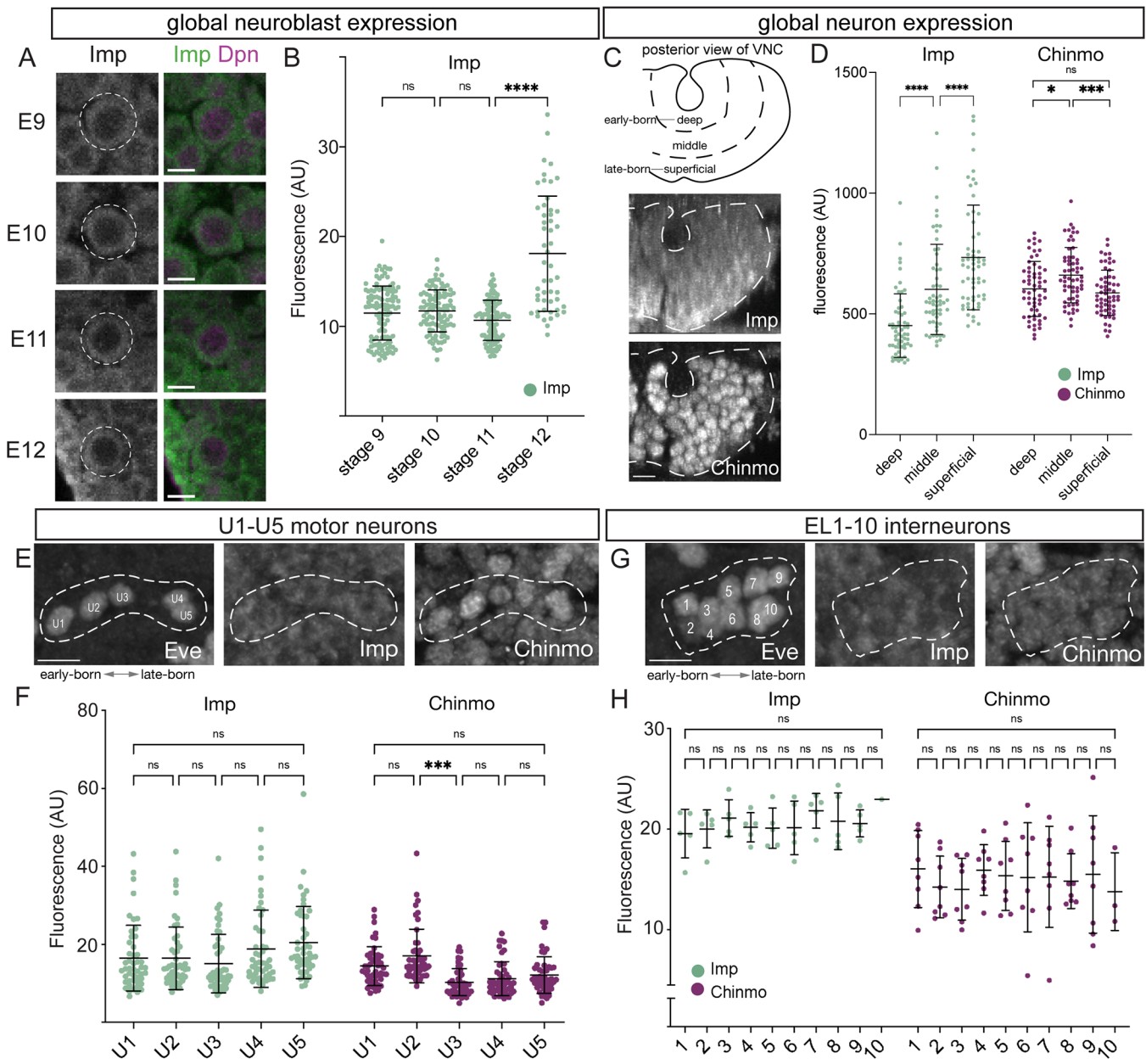

**Fig. 2. Imp forms a low-to-high temporal gradient in embryos.** (A,B) Imp forms a low-to-high gradient in aging embryonic neuroblasts. (A) Imp expression in Dpn+ neuroblasts at the indicated embryonic stages (left). Ventral view. Scale bars: 5 µm. (B) Quantification. *n*>40 for each stage. Imp expression increases significantly from stage 9-11 to stage 12 (stage 11 vs stage 12: *P*<0.0001, one-way ANOVA with Tukey's multiple comparisons test). (C,D) Imp forms a low-to-high gradient in aging embryonic neurons. (C) Imp and Chinmo expression in a cross-sectional (posterior) view, where older neurons are located in a deep layer and younger neurons are located in a more superficial layer. (D) Quantification. Imp is expressed in a low-to-high gradient (deep vs middle: *P*<0.0001; middle vs superficial: *P*<0.001, deep vs superficial: *P*<0.0001; one-way ANOVA with Tukey's multiple comparisons test). Chinmo is expressed but not in a gradient (deep vs superficial: *P*>0.05, one-way ANOVA with Tukey's multiple comparisons test). Scale bar: 5 µm. *n*>40 segments for each gene. (E,F) Imp and Chinmo do not form gradients in the young-old U1-U5 motor neurons, identified by expression of the Eve transcription factor. (E) Imp, Chinmo, and Eve expression in stage 16 embryos. Ventral view. Scale bar: 5 µm. (F) Quantification. No significant difference in Imp or Chinmo levels was detected between adjacent UMNs (*P*>0.05, one-way ANOVA with Tukey's multiple comparisons test), except that Chinmo levels are lower in U3 vs U2 (*P*<0.0001, Tukey's multiple comparisons test). *n*>20 for each neuron. (G,H) Imp and Chinmo do not form gradients in the young-old EL1-EL10 interneurons, identified by lateral expression of the Eve transcription factor. (G) Imp, Chinmo, and Eve expression in stage 16 embryos. Ventral view. Scale bar: 5 µm. (H) Quantification. No significant difference in Imp or Chinmo levels was detected between adjacent UMNs (*P*>0.05, one-way ANOVA with Tukey's multiple comparisons test). *n*≥3 for each neuron expect EL10 due to EL number variation. AU, arbitrary units; ns, not significant.

et al., 2022; Liu et al., 2015; Zhu et al., 2006). We wanted to know if these factors show the same or different modes of cross-regulation in the embryonic CNS. Above, we showed that wild-type Chinmo has relatively high expression and Imp has modest expression. Syp is not detected in neurons, although there is expression in cells outside

the CNS and sporadically in a subset of glia (Fig. 3A). We used an *Imp* mutant, *Imp⁷*, and observed loss of Chinmo (Fig. 3B, quantified in D). In *chinmo¹* mutant embryos, we observed loss of Imp expression and de-repression of Syp – unlike in *Imp⁷* mutants (Fig. 3C, quantified in D). Note that we did not assay *Syp* mutant

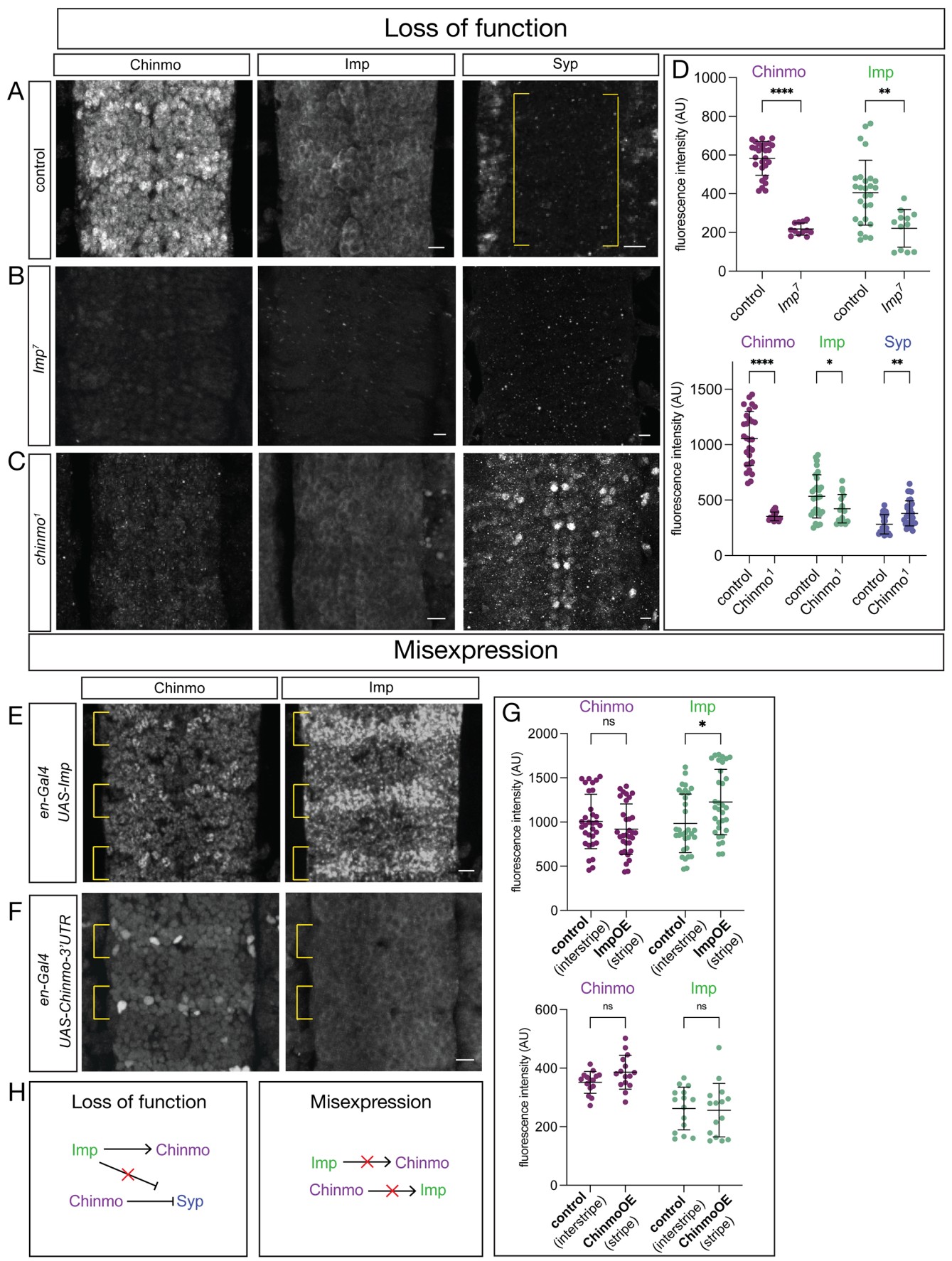

**Fig. 3.** See next page for legend.

**Fig. 3. Cross-regulation of Imp, Syp and Chinmo in the embryonic VNC.** (A) Control (*yw*) stained for Chinmo, Imp, and Syp. (B) *Imp*[7] homozygous mutant stained for Chinmo, Imp, and Syp. (C) *chinmo*[1] homozygous mutant stained for Chinmo, Imp, and Syp. (E) Imp overexpression (*en-gal4 UAS-Imp*) stained for Chinmo and Imp. (F) Chinmo overexpression (*en-gal4 UAS-chinmo-3′UTR*) stained for Chinmo and Imp. All panels show stage 16 embryos, ventral view, scale bars: 5 µm. (D) Quantification of Chinmo ($P<0.0001$, unpaired *t*-test) and Imp ($P<0.002$, unpaired *t*-test) expression in *Imp*[7] homozygous mutants compared to *Imp*[7] heterozygotes as control. Quantification in *chinmo*[1] mutants of Chinmo ($P<0.0001$, unpaired *t*-test), Imp ($P<0.05$, unpaired *t*-test and Syp ($P<0.002$, unpaired *t*-test), with *chinmo*[1] heterozygotes as control. (G) Quantification of Chinmo ($P>0.05$, ns, unpaired *t*-test) and Imp ($P<0.03$, unpaired-*t*-test) in control (En− interstripe) and ImpOE (En+ stripe). Quantification of Chinmo ($P>0.05$, ns, unpaired *t*-test) and Imp ($P>0.05$, ns, unpaired-*t*-test) in control (En− interstripe) and ChinmoOE (En+ stripe).

embryos because there is no expression of Syp in the wild-type CNS (Fig. 3A). In contrast, overexpression of Imp with *en-gal4* in NBs and neurons resulted in no change in Chinmo (Fig. 3E,F, quantified in G), despite the overexpression of Imp flattening the Imp temporal gradient (Fig. S3). Lastly, we assayed overexpression of Chinmo and found no change in levels of Imp expression (Fig. 3F, quantified in G). We conclude that (1) in contrast to larvae, Imp and Syp do not regulate each other in embryos, and (2) more specifically, loss of Imp and loss of Chinmo had different effects on Syp expression in embryos (Fig. 3H).

### Imp and Chinmo do not specify motor neuron molecular identity

To further understand the function of Imp and Chinmo function in embryos, we assayed their role in motor neuron molecular identity (this section) and motor neuron axon and dendrite targeting (next sections). We chose to analyze the U1-U5 motor neurons because they are well characterized for molecular identity (Isshiki et al., 2001) and axon/dendrite morphology (Seroka and Doe, 2019). In wild type, Eve is expressed in U1-U5, Zfh2 is expressed in U2-U5, Runt is expressed in U4 (high level) and U5 (low level), and Cut is weakly expressed in U3-U5 (Fig. 4A). We observed the same molecular identity in *Imp*[7] null mutants (Fig. 4B) and in *chinmo*[1] null mutants (Fig. 4C). We conclude that loss of Imp has no effect on the molecular identity of the U1-U5 motor neurons, although it is possible that maternal RNAs may obscure zygotic mutant phenotypes (see Discussion).

### Chinmo is required for motor neuron axon and dendrite targeting

Chinmo is not required for motor neuron molecular identity, but it may have a role in later events such as axon/dendrite morphogenesis, targeting, or connectivity (Marchetti and Tavosanis, 2017). We used *CQ-gal4* (expressed in newly post-mitotic U1-U5 neurons) to knock down Chinmo and assayed for targeting to the dorsal muscle field (U1/U2) or the lateral muscle field (U3-U5). In controls, we observed innervation of both dorsal and lateral muscle fields (Fig. 5A, quantified in C). In contrast, *chinmo* RNAi expressed in U1-U5 neurons resulted in frequent failure to innervate the dorsal muscles (Fig. 5B, quantified in C). We conclude that Chinmo is required for proper motor neuron-muscle connectivity.

Next, we used MCFO to specifically label individual U1-U5 motor neurons and confirmed that control U1/U2 motor neurons were bipolar and had dendritic arbors projecting away from the midline in newly hatched larvae (Fig. 5D, quantified in F). In contrast, *chinmo* RNAi expressed in U1-U5 neurons resulted in U1/U2 showing ectopic dendrite projections contacting the midline in newly hatched larvae (Fig. 5E, quantified in F). Similar abnormal

midline contacting was observed in the mono-polar U3-U5 motor neurons (Fig. 5G,H, quantified in I). We conclude that Chinmo functions to restrict dendrite outgrowth to the ipsilateral neuropil.

### Imp is required for motor neuron axon targeting

Imp is also not required for motor neuron molecular identity, but it may have a role in later events such as axon/dendrite morphogenesis, targeting, or connectivity (Munroe and Doe, 2023). We used *CQ-gal4* to knock down Imp in U1-U5 neurons and assayed for targeting to the dorsal muscle field (U1/U2) or the lateral muscle field (U3-U5). In controls, we observed innervation of both dorsal and lateral muscle fields (Fig. 6A, quantified in C). In contrast, expression of *Imp* RNAi in U1-U5 neurons resulted in frequent failure to innervate the dorsal muscles (Fig. 6B, quantified in C). We conclude that Imp is required for proper motor neuron-muscle connectivity.

Next, we used MCFO to specifically label individual U1-U5 motor neurons. Similar to *chinmo* RNAi, we found that *Imp* RNAi resulted in ectopic U1-U2 dendrite projections crossing the midline (Fig. 6D,E, quantified in F). We did not observe dendrite phenotypes in U3-U5 motor neurons (Fig. 6G,H, quantified in I). We conclude that Imp is required to prevent ectopic motor neuron dendritic targeting. The relative weakness of the phenotype compared to the *chinmo* RNAi phenotype could be due to persistent maternal Imp RNA, or a lack of Imp function in shaping U3-U5 dendrite morphology (see Discussion). We conclude that Imp may also functions to restrict dendrite outgrowth to the ipsilateral neuropil specifically in early-born U1-U2 neurons.

## DISCUSSION
### Cross-regulation

We find many contrasting cross-regulatory interactions of Imp, Chinmo, and Syp during embryonic CNS development compared to larval CNS development (Fig. 7). We observed the following differences. (1) Whereas larval NBs express opposing gradients of Imp and Syp (Liu et al., 2015), in embryos only Imp is expressed while Syp is undetectable, and thus embryos have no role for Syp in the CNS. (2) Whereas larval NBs show Imp activating Chinmo but not the opposite (Zhu et al., 2006), in embryos both Imp and Chinmo positively regulate each other. (3) Whereas larval neurons do not show Chinmo repressing Syp (Liu et al., 2015), in embryos Chinmo clearly represses Syp. (4) Whereas larval NBs show pan-neuronal expression of the mid (Mamo) or late (E93, Broad) temporal factors (Ren et al., 2017; Syed et al., 2017), in embryos Mamo and E93 are not expressed in the CNS, and Broad is only detected in a small subset of neurons, indicating that they play no role in regulating Imp or Chinmo.

Syp is an RNA-binding protein and has been shown to bind *chinmo* mRNA to repress Chinmo expression (Liu et al., 2015; Zhu et al., 2006). Interestingly, we found that Chinmo is required to repress Syp expression in the embryonic CNS; however, knockdown of *Imp* has no effect on Syp levels, consistent with our result showing that Syp is not expressed in the embryonic CNS. Note that *Imp* and *chinmo* RNA are maternally provided (https://insitu.fruitfly.org/cgi-bin/ex/insitu.pl), and thus maternal expression of *Imp* in zygotic *Imp*[7] mutants may be sufficient to repress Syp expression.

### Neuronal identity

Previous work has shown that *Imp* is required in larvae for Kenyon cell specification: knockdown of *Imp* or *Syp* alters the ratio of early-born versus late-born neuronal identity (Hamid et al., 2024; Liu et al., 2015). In contrast, our results show that *Imp* and *chinmo* have no role in motor neuron specification at the molecular marker level.

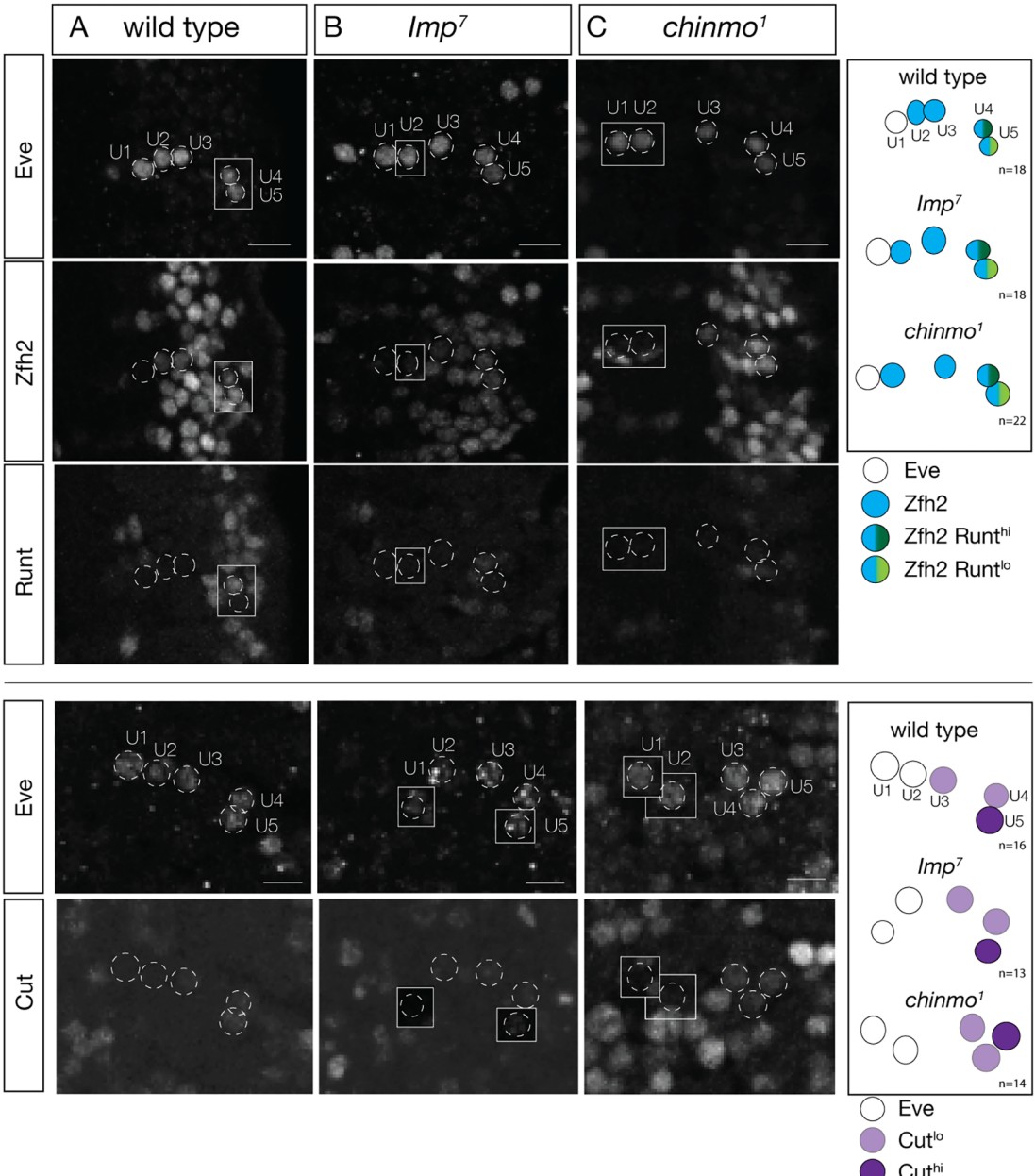

**Fig. 4. Imp and Chinmo are not required for motor neuron identity.** (A) Control (*yw*); molecular markers show wild-type U1-U5 motor neuron identity. Eve, U1-U5; Zfh2, U2 (low), U3-U5; Runt, U4 (high)-U5 (low); Cut (U4, U5). Scale bars: 5 µm. (B) *Imp⁷* homozygous mutant; molecular markers show wild-type U1-U5 motor neuron identity. Eve, U1-U5; Zfh2, U2-U5; Runt, U4 (high)-U5 (low); Cut (U3, U4, U5). Scale bars: 5 µm. (C) *chinmo¹* homozygous mutant; molecular markers show wild-type U1-U5 motor neuron identity. Eve, U1-U5; Zfh2, U2-U5; Runt, U4 (high)-U5 (low); Cut (U3, U4, U5). Scale bars: 5 µm.

Similarly, at larval stages, loss of *Imp* does not alter expression of pMad, a marker of motor neurons (Boylan et al., 2008), suggesting that *Imp* is also dispensable in larvae for specification of motor neuron molecular identity.

### Timing

There is a major difference in time scale between embryonic and larval neurogenesis: embryonic neurogenesis is completed in less than 1 day, whereas larval neurogenesis lasts 5 days. The shorter time of embryonic neurogenesis may require a more 'hard-wired' mechanism such as a TTF cascade that switches TTF expression approximately every hour (Pollington et al., 2023). In contrast, longer larval neurogenesis may provide time to generate and

respond to gradients of Imp and Syp RNA-binding proteins (Islam and Erclik, 2022). We found that overexpression of Hb in NBs, which stalls the TTF cascade (Isshiki et al., 2001), does not alter the Imp temporal gradient in post-mitotic neurons. Additionally, overexpression of Imp, creating levels similar to Imp expression in late-born neurons, does not alter the timing of Hb expression in NBs. These results suggest that Imp/Chinmo and the TTF cascade function in separate pathways to guide neuronal development.

### Axon/dendrite morphology

In larval motor neurons, *Imp* mutants display reduced motor neuron bouton number at the neuromuscular junction (Boylan et al., 2008) and failure in axon/dendrite targeting within the central complex

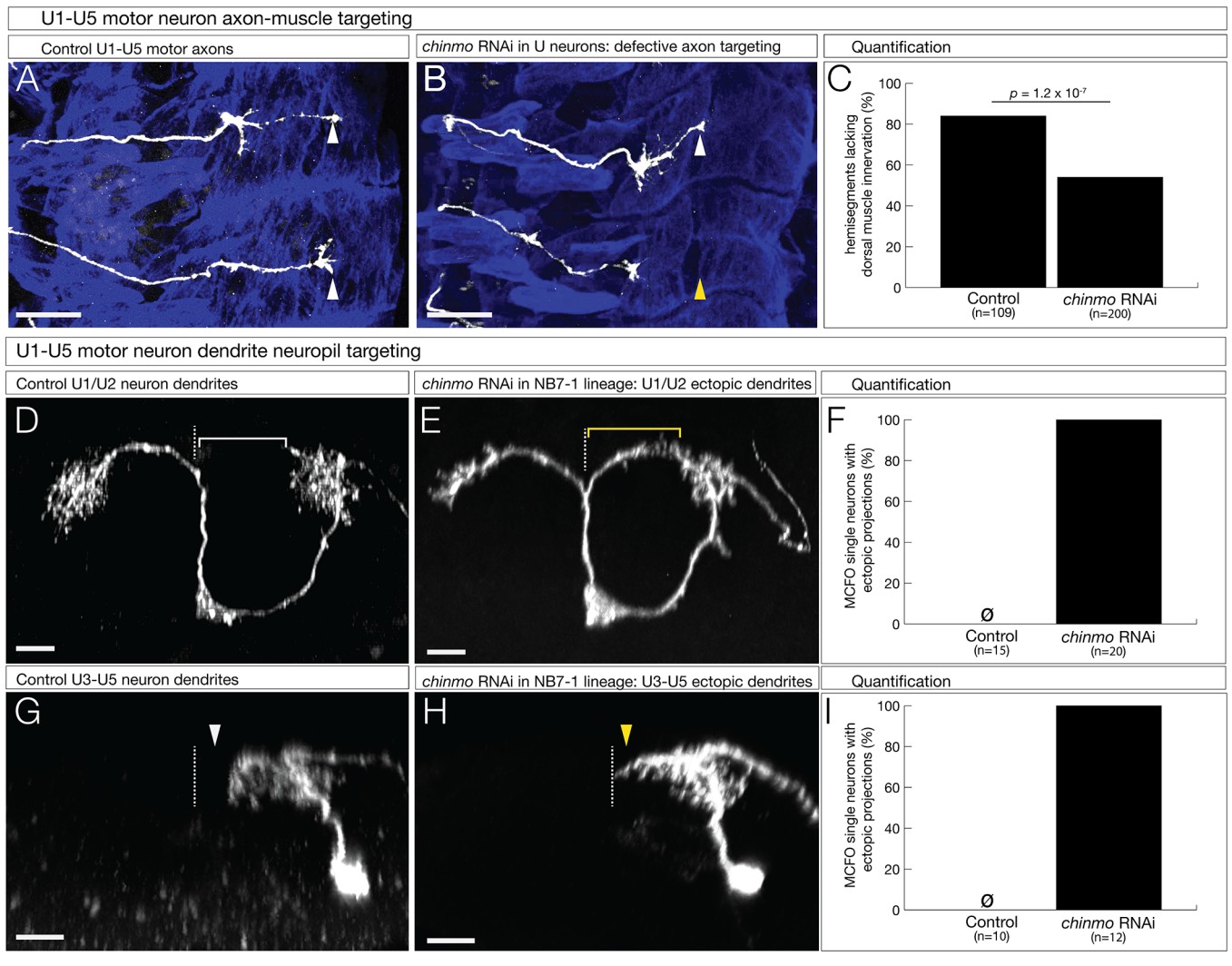

**Fig. 5. Chinmo is required for motor neuron axon and dendrite targeting.** (A-C) Axon targeting. U1-U5 motor neurons were labeled with GFP; lateral view, anterior up, dorsal to the right. Genetics: *CQ-Gal4, UAS-myr-GFP*, assayed at late stage 17. (A) Controls (*+UAS-LacZ*) have extension of motor neurons to the dorsal muscle field. Scale bar: 20 µm. (B) In *chinmo* mutant (*+UAS-chinmo-RNAi*) motor neurons, their axons fail to project to the dorsal muscle field. Scale bar: 20 µm. (C) Quantitation of hemisegments lacking dorsal innervation. (D-F) Dendrite targeting of U1-U2 motor neurons. Multicolor flip out (MCFO) to obtain single motor neuron dendrites. Posterior view; midline, dashed line. Genetics: *NB7-1-Gal4, R57C10-Flp, UAS-MCFO,* assayed in newly hatched larvae. (D) Control (*+UAS-LacZ*) U1-U2 motor neuron. Note lack of midline crossing (white bracket). Scale bar: 5 µm. (E) *chinmo* mutant (*+UAS-chinmo-RNAi*) showing ectopic dendrite arbors crossing the midline (yellow bracket). Scale bar: 5 µm. (F) Quantitation of motor neuron with dendrites crossing the midline. (G-I) Dendrite targeting of U3-U5 motor neurons. MCFO to obtain single motor neuron dendrites. Posterior view; midline, dashed line. Genetics: *NB7-1-Gal4, R57C10-Flp, UAS-MCFO,* assayed in newly hatched larvae. (G) Control (*+UAS-LacZ*) U3-U5 motor neuron. Note lack of midline contact (white arrowhead). Scale bar: 5 µm. (H) *chinmo* mutant (*+UAS-chinmo-RNAi*) showing motor neuron dendrite contacting the midline (yellow arrowhead). Scale bar: 5 µm. (I) Quantitation of motor neurons with dendrites contacting the midline.

(Munroe and Doe, 2023). Consistent with larval observations, we find that *Imp* and *Chinmo* mutants both show failure of motor axons to reach their correct body wall muscle target, and dendrites reaching or crossing the midline. Previous work imaging movement of Imp: GFP in motor axons showed that Imp is trafficked bidirectionally (Boylan et al., 2008); this movement may be defective in *Imp* mutants and explain the reduced bouton formation in larval motor neurons. It seems likely that Imp is also moving bidirectionally in embryonic motor neurons; this would be an interesting question to explore in the future. Our finding that Imp and Chinmo have similar roles in motor neuron axon targeting suggests that they are acting in the same pathway. Imp is an RNA-binding protein, and its RNA cargo have been defined in larval NBs, where Imp plays a role in regulation of proliferation (Samuels et al., 2020; Yang et al., 2017); these RNAs are also good candidates for a role in embryonic dendrite and axon

targeting. One way Chinmo may regulate neurite targeting is through repression of Syp expression. It was previously reported that *Syp*, along with muscle-specific protein *msp300*, which enables actin binding activity, are required for synapse formation (Titlow et al., 2020). Additionally, Syp is required for synaptic plasticity in motor neurons (Titlow et al., 2020). Derepression of Syp in *chinmo[1]* mutants may promote additional synapse formation or synaptic plasticity in motor neuron dendrites leading to ectopic neurite formation. Moreover, it is possible Chinmo has roles in other neuronal cell types beyond motor neurons. Further work to understand the targets of Imp, Chinmo, and Syp that lead to proper neurite targeting will be an important task for the future.

Might these defects in axon/dendrite targeting prevent normal locomotor behavior? We note that defects are only observed in a subset of segments, and the experiment generates loss of Imp or Chinmo only

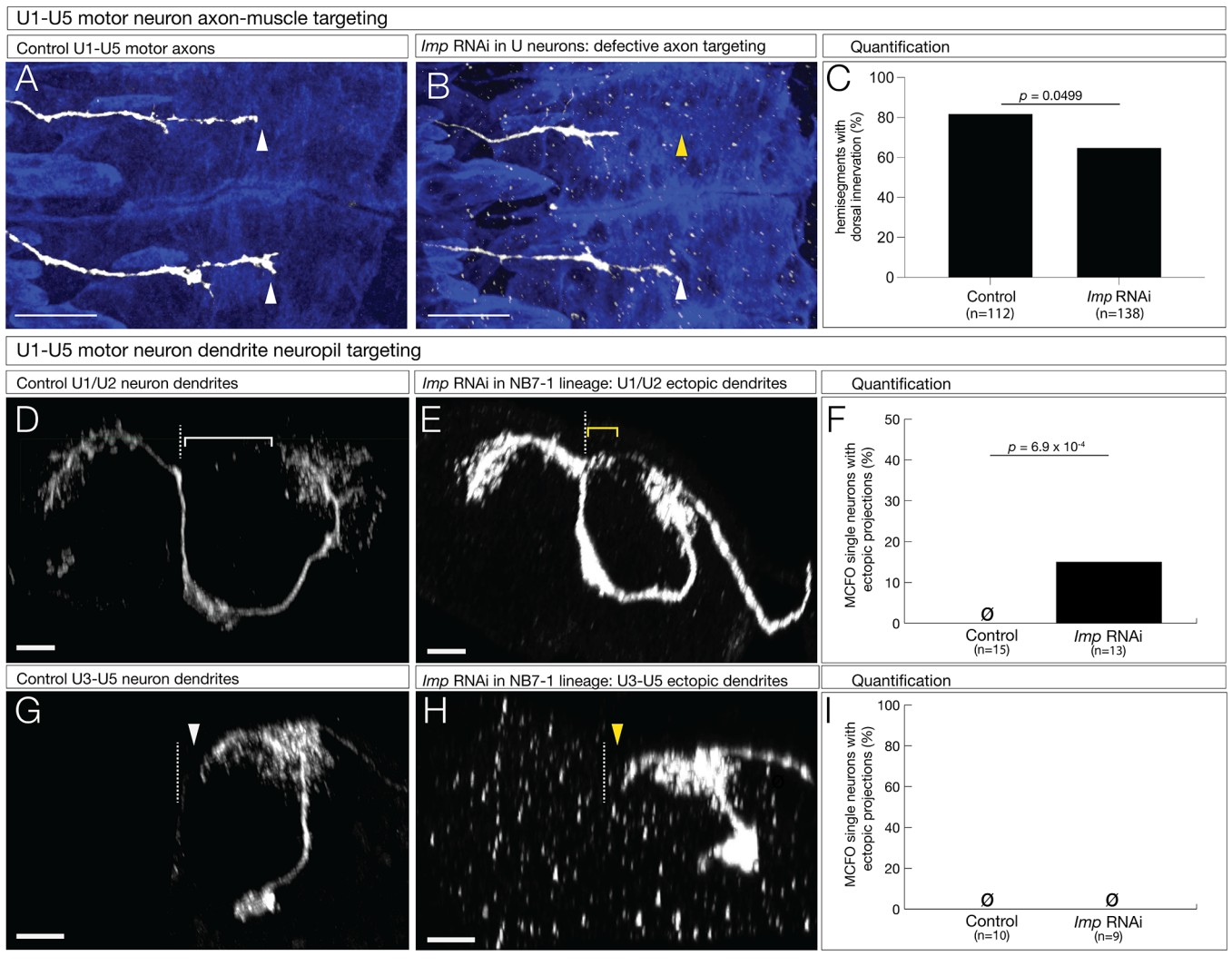

**Fig. 6. Imp is required for motor neuron axon and dendrite targeting.** (A-C) Axon targeting. U1-U5 motor neurons were labeled with GFP; lateral view, anterior up, dorsal to the right. Genetics: *CQ-Gal4, UAS-myr-GFP*, assayed at late stage 17. (A) Controls (+*UAS-LacZ*) have extension of motor neurons to the dorsal muscle field. Scale bar: 20 µm. (B) In *Imp* mutant (+*UAS-Imp-RNAi*) motor neurons, their axons fail to project to the dorsal muscle field. Scale bar: 20 µm. (C) Quantitation of hemisegments lacking dorsal innervation. (D-F) Dendrite targeting of U1-U2 motor neurons. MCFO to obtain single motor neuron dendrites. Posterior view; midline, dashed line. Genetics: *NB7-1-Gal4, R57C10-Flp, UAS-MCFO*, assayed in newly hatched larvae. (D) Control (+*UAS-LacZ*) U1-U2 motor neuron. Note lack of midline crossing (white bracket). Scale bar: 5 µm. (E) *Imp* mutant (+*UAS-Imp-RNAi*) showing ectopic dendrite arbors crossing the midline (yellow bracket). Scale bar: 5 µm. (F) Quantitation of motor neuron with dendrites crossing the midline. (G-I) Dendrite targeting of U3-U5 motor neurons. MCFO to obtain single motor neuron dendrites. Posterior view; midline, dashed line. Genetics: *NB7-1-Gal4, R57C10-Flp, UAS-MCFO*, assayed in newly hatched larvae. (G) Control (+*UAS-LacZ*) U3-U5 motor neuron. Note lack of midline contact (white arrowhead). Scale bar: 5 µm. (H) *Imp* RNAi (+*UAS-Imp-RNAi*) in U3-U5 showing no phenotype (yellow arrowhead). Scale bar: 5 µm. (I) Quantitation of motor neurons with dendrites contacting the midline.

in the five U motor neurons and does not affect other MNs that innervate the dorsal muscles (e.g. RP2). Thus, this highly targeted manipulation is unlikely to generate an organismal phenotype.

## MATERIALS AND METHODS
### Fly stocks
We used *Drosophila melanogaster* males and females. We used the following fly stocks. *Imp[7]* (Syed Lab), *chinmo[1]* [Bloomington *Drosophila* Stock Center (BDSC):59969], *en-gal4* (BDSC:99568), *UAS-Imp* (BDSC:93386), *UAS-chinmo-3′UTR* (BDSC:44388), *UAS-hb* (Tran et al., 2010), *UAS-Imp-RNAi* (BDSC:38219), *UAS-chinmo-RNAi* (BDSC:33638), CQ-gal4 (Hirono et al., 2017; BDSC:7466), *NB7-1-gal4* (Seroka and Doe, 2019).

### Antibody staining
For embryonic CNS imaging, embryos were transferred from apple caps into collection baskets and rinsed with dH$_2$O. Embryos were dechorionated in

100% bleach (Clorox, Oakland, CA, USA) for 4 min with gentle agitation. Dechorionated embryos were rinsed with dH$_2$O for 30 s. Embryos were fixed 20 min in 2 ml Eppendorf tubes containing equal volumes of heptane (Fisher Chemical, Eugene, OR, USA; H3505K-4) and 4% paraformaldehyde diluted in PEM [100 mM PIPES pH6.95 (Sigma-Aldrich, St Louis, MO, USA), 2 mM EDTA pH8.0 (Sigma-Aldrich) and 1 mM MgSO$_4$ (Sigma-Aldrich). The lower fix layer was removed, and an equal volume of methanol was added to each tube. Tubes were then subject to vigorous agitation for 1 min in a step required for removing the vitelline membrane. Nearly all liquid was removed from the tubes, leaving the embryos. Embryos were rinsed in methanol (Fisher Chemical, Lot# 206197, A412P-4) three times and stored at −20°C. Embryos were washed three times for 5 min with rocking in 0.1% PBST (1×PBS/0.1% Triton-X 100). PBST was removed and embryos were blocked with 5% normal donkey serum (Jackson ImmunoResearch, West Grove, PA, USA) in PBST for 30 min at room temperature with rocking. PBST was removed, and antibody mixes in PBST were added and rocked overnight at 4°C. Primary antibody mixes were removed, and embryos were washed for >15 min three

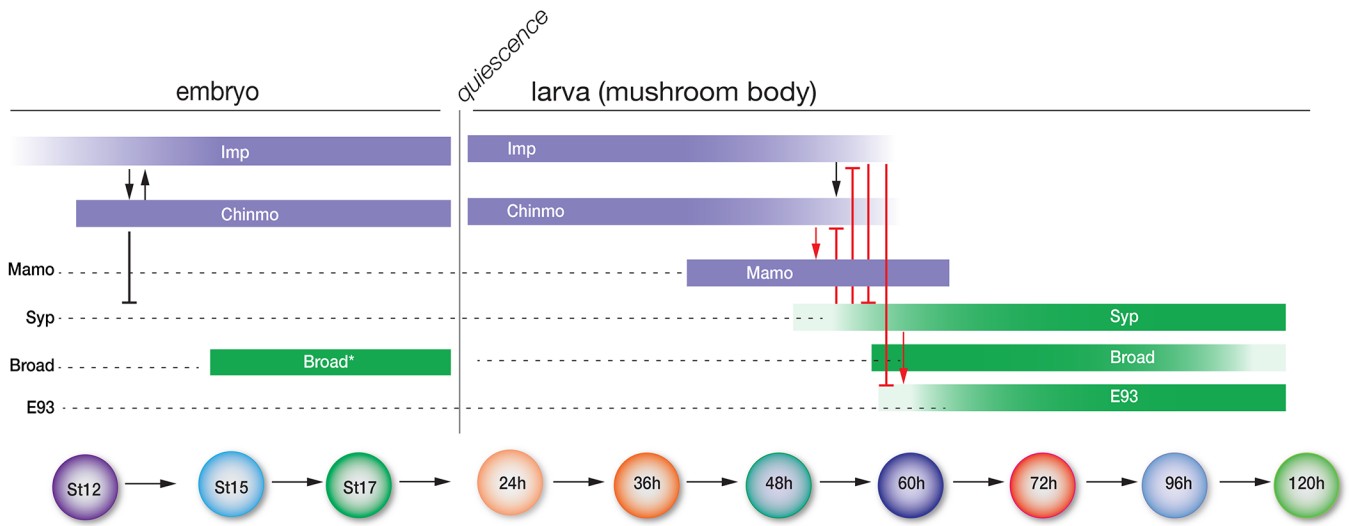

**Fig. 7. Schematic.** Known roles of the indicated larval temporal factors over time are shown on the right; expression and cross-regulation of the indicated temporal factors in the embryonic VNC are shown on the left. St, embryonic stage; h, hours after larval hatch. Timeline not to scale. Arrows, positive regulation; T bars, repressive regulation. Red text indicates larval-specific gene regulation mechanisms. 'Broad*' indicates scattered neuronal expression, not pan-neuronal.

times in PBST with rocking. PBST was removed, and secondary antibody diluted in PBST was added. Embryos were rocked at room temperature for 2 h or rocked overnight at 4°C. Embryos were washed for >15 min three times in PBST with rocking.

After washing off secondary antibody, embryos were washed three times in PBS then mounted in lysine coverslips and dehydrated in an ethanol series (30%, 50%, 70%, 90%, 100%). Embryos were washed an additional time in 100% ethanol. Next, embryos were washed two times in xylenes (Sigma-Aldrich), then mounted in DPX mounting medium (Sigma-Aldrich) and dried at room temperature for 2 days before imaging.

### Axon/dendrite experiments
For analysis of axon targeting, embryos were collected over a 24-h window then fixed in 4% paraformaldehyde, and stage 17 embryos were immunostained. Embryo staging was done according to gut morphology, ensuring that both controls and experimentals were at the same age (late stage 17). For imaging, embryos were transferred into 50% glycerol for 20 min or until embryos have settled at the bottom of the tube. The 50% glycerol was removed, and 90% glycerol was added. Embryos were left at room temperature overnight to let them fully settle to the bottom of the tube before imaging.

For analysis of dendrite targeting, embryos were collected over a 24-h window then aged for 24 h. Freshly hatched larval brains were dissected in PBS or HL3.1 (Feng et al., 2004), fixed in 4% paraformaldehyde, and mounted on lysine coverslips with DPX. Brains were immunostained on coverslips.

### Primary and secondary antibodies
Primary antibodies used were as follows: chicken anti-GFP, 1:1000 (Aves Labs, Davis, CA, USA); rabbit anti-Imp, 1:500 (Doe laboratory); rabbit anti-Syp, 1:1000 (Desplan laboratory, NYU, New York, NY, USA); rat anti-Deadpan, 1:20 (Doe laboratory); mouse anti-Hunchback, 1:200 (Abcam, Eugene, OR, USA); mouse anti-Eve, 1:100 [Developmental Studies Hybridoma Bank (DSHB), Iowa City, IA, USA]; guinea pig anti-Chinmo, 1:200 (Desplan laboratory); rat anti-Zfh2, 1:250 (Doe laboratory); mouse anti-Broad, 1:20 (DSHB); guinea pig anti-Mamo, 1:200 (Desplan laboratory); guinea pig anti-E93, 1:500 (Doe laboratory); mouse anti-En, 5 µg/ml (DSHB); mouse anti-Eve [2B8], 5 µg/ml (DSHB); rabbit anti-Eve, 1:250 (Doe laboratory); rabbit anti-Hb, 1:200 (Tran and Doe, 2008); guinea pig anti-Runt, 1:1000 (Sullivan et al., 2019); rat anti-Tm1 [MAC141], 1:500 (Abcam, Waltham, MA, USA); mouse anti-HA [901513] (BioLegend, San Diego, CA, USA); chicken anti-V5 [A190-218A] (Bethyl Laboratories, Boston, MA, USA); rat anti-Flag [NBP1-06712] (Novus Biologicals, Centennial, CO, USA); rat anti-OLLAS [NBP1-06713] (Novus).

Secondary antibodies used were DyLight 405, AlexaFluor 488, rhodamine Red™-X, AlexaFluor 555, or Alexa Fluor 647-conjugated AffiniPure™ donkey anti-IgG (Jackson ImmunoResearch). The samples were mounted in 90% glycerol with Vectashield (Vector Laboratories, Burlingame, CA, USA) or DPX (Sigma-Aldrich).

### Confocal microscopy
Images were captured with a Zeiss LSM800, LSM 900 or LSM 900-Airyscan2 laser scanning confocal microscope with a z-resolution of 0.25 µm (Carl Zeiss, Oberkochen, Germany) equipped with an Axio Imager.Z2 microscope. A 40×/1.40 NA Oil Plan-Apochromat DIC m27 objective lens and a 63×/1.40 Oil Plan-Apochromat DIC m27 objective lens and GaAsP photomultiplier tubes were used. Software program was Zen 2.3 (blue edition) (Carl Zeiss). Images were processed using the open-source software FIJI or Imaris (Oxford Instruments, Abingdon, UK). Figures were assembled in Illustrator (Adobe, San Jose, CA, USA). For each independent experiment, all samples were acquired using identical acquisition parameters.

### Statistical analyses
Statistics were computed using Prism (GraphPad, Boston, MA, USA). One-way ANOVA with Tukey's multiple comparison test was used in Fig. 2. Unpaired $t$-test was used for comparison of wild-type and experimental conditions in Figs 3-6. $P$-value scale defined by Prism: ns, not significant ($P>0.05$); *$P<0.033$, **$P<0.0021$, ***$P<0.0002$, ****$P<0.0001$. $P$-values are reported in the figures. Plots were generated using Prism (GraphPad).

### Figure production
Images for figures were processed in FIJI. Figures were assembled in Adobe Illustrator. Any changes in brightness or contrast were applied to the entire image.

### Acknowledgements
We thank Heather Pollington, Tory Herman, and Megan Radler for constructive comments on the manuscript, and Megan Radler and Josmarie Graciani for assistance with brain dissections. Stocks obtained from the BDSC (NIH P40OD018537) were used in this study. Flybase was used extensively in this work; FlyBase's data and annotations are available with a CC BY 4.0 license.

### Competing interests
The authors declare no competing or financial interests.

### Author contributions
Conceptualization: K.H.F., S.-L.L., C.Q.D.; Data curation: K.H.F., S.-L.L., C.Q.D.; Formal analysis: K.H.F., S.-L.L.; Funding acquisition: K.H.F., C.Q.D.; Investigation:

K.H.F., S.-L.L.; Methodology: K.H.F., S.-L.L.; Project administration: S.-L.L., C.Q.D.; Resources: K.H.F., S.-L.L.; Supervision: S.-L.L., C.Q.D.; Validation: K.H.F., S.-L.L.; Visualization: K.H.F., S.-L.L.; Writing – original draft: K.H.F., C.Q.D.; Writing – review & editing: K.H.F., S.-L.L., C.Q.D.

## Funding

Funding was provided by Howard Hughes Medical Institute (HHMI; C.Q.D., S.-L.L.) and National Institutes of Health 5F31HD108945-02 (K.H.F.). Open Access funding provided by HHMI. Deposited in PMC for immediate release.

## Data availability

All relevant data and details of resources can be found within the article and its supplementary information.

## Peer review history

The peer review history is available online at https://journals.biologists.com/bio/lookup/doi/10.1242/bio.062105.reviewer-comments.pdf

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
