## [Peer Review File · Biology Open]

Imp and Chinmo are required for embryonic motor neuron axon and dendrite targeting

Katie H. Fisher, Sen-Lin Lai and Chris Doe
DOI: 10.1242/bio.062105

Editor: Tristan Rodríguez

Review timeline

Original submission:	6 June 2025
Editorial decision:	16 June 2025
First revision received:	16 June 2025
Accepted:	18 June 2025

Original submission

First decision letter

MS ID#: bio.062105

MS TITLE: Imp and Chinmo are required for embryonic motor neuron axon and dendrite targeting

AUTHORS: Katie H. Fisher; Sen-Lin Lai; Chris Doe

I have now reached a decision on the above manuscript.

The reviewer reports are shown at the bottom of this email or can be accessed, together with a copy of this decision letter, by going to:

As you will see, the reviewers gave favourable reports, but raised some critical points that will require amendments to your manuscript. I hope that you will be able to carry these out, because we would like to be able to accept your paper.

At this stage, we also ask you to ensure your manuscript complies with our formatting guidelines' please see our manuscript preparation guidelines for details. Provided you are able to fully address the referees' comments, we are positive about publication of your paper (we accept over 95% of revision submissions) and therefore hope you won't mind any extra work involved in reformatting your manuscript at this point.

Please ensure that you clearly highlight all changes made in the revised manuscript. Please avoid using 'Tracked changes' in Word files as these are lost in PDF conversion.

I should be grateful if you would also provide a point-by-point response detailing how you have dealt with the points raised by the reviewers in the 'Response to Reviewers' box. Please attend to all of the reviewers' comments. If you do not agree with any of their criticisms or suggestions please explain clearly why this is so.

Reviewer 1

Comments to Author

The manuscript by Fisher et al. addresses the contribution of the RNA binding proteins Imp and Syp, as well as their target transcription factor Chinmo, during the development of the embryonic CNS. They also analyze the expression of different larval temporal transcription factors such as Broad, E93 and Mamo. The function of these genes was previously characterized in the larval CNS, and the authors find compelling evidence to suggest that they are also used in embryos. They show a variety of differences in the regulatory mechanisms and requirements for these genes, defining Chinmo expression exclusively in embryonic neurons, and identifying a function of Imp and Chinmo for the establishment of correct axonal and dendrite projections of embryonic motor neurons. All together this is a straight forward manuscript, where the expression and phenotypic data are well described and quantified, and the description of the differences in regulatory mechanisms and functional requirements between larval and embryonic neural development are exhaustively and correctly described.

The authors find very consistent alterations in axon and dendrite morphology and projections in motor neurons, and this leads me to the only additional question the authors might like to address related to the overall functional consequences of these alterations in the embryonic muscular system. To what extent the observed morphological alterations in neurite targeting impact the activity of the neuromuscular system? Do embryonic movements such as coordinated crawling sequences are affected in Imp and Chinmo mutants?

Reviewer 2

Comments to Author

In this manuscript, Chris Doe and colleagues use *Drosophila* as model system to analyze the role in the development of the embryonic central nervous system of the so-called "temporal" transcriptional factors and RNA-binding proteins (Imp, Syp, Chinmo, Mamo, Broad, E93) that generate neuronal diversity at later stages of fly development to build the final adult brain. Adults combine expression analysis and functional genetics to conclude that, in contrast to the larval CNS, these molecular elements are not directly required to generate neuronal identity. The mechanisms of inter-regulation are different and they unravel new roles of these factors (Chimo and Imp) in axon/dendrite morphology. The paper is well-written, figures self-explanatory and the main message becomes clear in all the sections.

Minor issues:

- (1) When talking about gradients, I guess authors are intending to talk about "temporal" gradients (and not physical ones), right? Author should make this clear in the description of the known data in the larva and new data in the embryo.
- (2) Line 12: "Imp expression levels are consistent in NBs until stage 12..." unclear what consistent means....perhaps "constant"...
- (3) Line 113: "...we used en-gal4 to misexpress UAS-hb which is known to stall the TTF cascade at the first TTF..." Authors should explain where en-gal4 is expressed...
- (4) Lines 122-123: Unclear sentence....grammar is not correct and meaning difficult to extract.
- (5) Authors might want to include the newly discovered functions of Imp and Chimno in the embryo in Figure 7 and reinforce the role of these and other factors in generating neuronal diversity.
- (6) Acknowledgement to Flybase would be fair.

Reviewer's Responses to Questions

Experimental quality

Does each figure have the proper controls?

If 'No', please indicate reasons in Comments for Author box below.

Reviewer #1:

- Yes

Reviewer #2:

- Yes

Were the data analyzed using appropriate statistical tests?

If 'No', please indicate reasons in Comments for Author box below.

Reviewer #1:

- Yes

Reviewer #2:

- Yes

Reproducibility

Were experiments performed using adequate number of biological replicates?

If 'No', please indicate reasons in Comments for Author box below.

Reviewer #1:

- Yes

Reviewer #2:

- Yes

Does the methods section provide sufficient detail to permit reproducibility?

If 'No', please indicate reasons in Comments for Author box below.

Reviewer #1:

- Yes

Reviewer #2:

- Yes

Completeness

Are the manuscript's conclusions supported by the data?

If 'No', please indicate reasons in Comments for Author box below.

Reviewer #1:

- Yes

Reviewer #2:

- Yes

Scholarship

Do the authors cite and discuss the merits of data that would argue for and against their conclusion?

If 'No', please indicate reasons in Comments for Author box below.

Reviewer #1:

- Yes

Reviewer #2:

- Yes

Does the manuscript title & abstract accurately reflect the contents of the manuscript, without hyperbole?

If 'No', please indicate reasons in Comments for Author box below.

Reviewer #1:

- Yes

Reviewer #2:

- Yes

First revision

Reviewer response

Reviewer 1: The manuscript by Fisher et al. addresses the contribution of the RNA binding proteins Imp and Syp, as well as their target transcription factor Chinmo, during the development of the embryonic CNS. They also analyze the expression of different larval temporal transcription factors such as Broad, E93 and Mamo. The function of these genes was previously characterized in the larval CNS, and the authors find compelling evidence to suggest that they are also used in embryos. They show a variety of differences in the regulatory mechanisms and requirements for these genes, defining Chinmo expression exclusively in embryonic neurons, and identifying a function of Imp and Chinmo for the establishment of correct axonal and dendrite projections of embryonic motor neurons. All together this is a straight forward manuscript, where the expression and phenotypic data are well described and quantified, and the description of the differences in regulatory mechanisms and functional requirements between larval and embryonic neural development are exhaustively and correctly described.

We thank the reviewer for their positive comments.

The authors find very consistent alterations in axon and dendrite morphology and projections in motor neurons, and this leads me to the only additional question the authors might like to address related to the overall functional consequences of these alterations in the embryonic muscular system. To what extent the observed morphological alterations in neurite targeting impact the

activity of the neuromuscular system? Do embryonic movements such as coordinated crawling sequences are affected in *Imp* and *Chinmo* mutants?

The defects are only observed in a subset of segments, and the experiment generates loss of *Imp* or *Chinmo* in the five U motor neurons, and does not affect other MNs that innervate the dorsal muscles (e.g. RP2). Thus, this highly targeted manipulation is unlikely to generate an organismal phenotype. We have added this to the discussion.

Reviewer 2: In this manuscript, Chris Doe and colleagues use *Drosophila* as model system to analyze the role in the development of the embryonic central nervous system of the so-called "temporal" transcriptional factors and RNA-binding proteins (*Imp*, *Syp*, *Chinmo*, *Mamo*, *Broad*, *E93*) that generate neuronal diversity at later stages of fly development to build the final adult brain. Adults combine expression analysis and functional genetics to conclude that, in contrast to the larval CNS, these molecular elements are not directly required to generate neuronal identity. The mechanisms of inter-regulation are different and they unravel new roles of these factors (*Chinmo* and *Imp*) in axon/dendrite morphology. The paper is well-written, figures self-explanatory and the main message becomes clear in all the sections.

We thank the reviewer for their positive comments.

Minor issues:

(1) When talking about gradients, I guess authors are intending to talk about "temporal" gradients (and not physical ones), right? Author should make this clear in the description of the known data in the larva and new data in the embryo.

That is correct, we have now clarified this point in the text. We changed the term "gradient" to "temporal gradient" where it occurs in the text (29 locations).

(2) Line 12: "*Imp* expression levels are consistent in NBs until stage 12..." unclear what consistent means....perhaps "constant"....?

Thanks, we have fixed this in the revised text. From "consistent" to "similar"

(3) Line 113: "...we used *en-gal4* to misexpress *UAS-hb* which is known to stall the TTF cascade at the first TTF..." Authors should explain where *en-gal4* is expressed...

Thanks, we now say: "To determine if the TTF cascade generates the *Imp* temporal gradient, we used *en-gal4*, which is expressed in stripes within the posterior domain of each segment, to misexpress *UAS-hb* -- which is known to stall the TTF cascade at the first TTF (Isshiki et al., 2001; Pollington and Doe, 2024; Tran and Doe, 2008) -- and assayed *Imp* levels."

(4) Lines 122-123: Unclear sentence....grammar is not correct and meaning difficult to extract.

Thanks, this was a broken sentence; we have revised the text fix the error.

(5) Authors might want to include the newly discovered functions of *Imp* and *Chinmo* in the embryo in Figure 7 and reinforce the role of these and other factors in generating neuronal diversity.

We are sorry that we were unclear. *Imp* and *Chinmo* have no effect on molecular neuronal identity. Figure 7 is designed to illustrate the differences in cross-regulation in embryos versus larvae (and there are many differences). We have highlighted text summarizing the role of *Imp* and *Chinmo* in axon and dendrite targeting: " Consistent with larval observations, we find that *Imp* and *Chinmo* mutants both show failure of motor axons to reach their correct body wall muscle target, and dendrites to remain off the midline. "

(6) Acknowledgement to Flybase would be fair.

We agree! And we now acknowledge Flybase. " Flybase (<https://flybase.org/>) was used extensively in this work; FlyBase's data and annotations are available with a CC BY 4.0 license. "

Second decision letter

MS ID#: bio.062105R1

MS TITLE: Imp and Chinmo are required for embryonic motor neuron axon and dendrite targeting

AUTHORS: Katie H. Fisher; Sen-Lin Lai; Chris Doe

I am happy to tell you that your manuscript has been accepted for publication in Biology Open, pending our standard publication integrity checks. It was accepted on 18 Jun 2025.